# **Resolution analysis of joint inversion of seismic receiver function and surface wave dispersion curves in the "13 BB Star" experiment**

Kajetan Chrapkiewicz<sup>1,2</sup>, Monika Wilde-Piórko<sup>1</sup>, Marcin Polkowski<sup>1</sup>, and Marek Grad<sup>1</sup> <sup>1</sup>University of Warsaw, Faculty of Physics, Institute of Geophysics, Warsaw, Poland <sup>2</sup>currently at Imperial College London, Department of Earth Science and Engineering, London, United Kingdom *Correspondence to:* Kajetan Chrapkiewicz (k.chrapkiewicz17@imperial.ac.uk)

**Abstract.** Joint inversion of Rayleigh wave phase velocity dispersion and P receiver function has been applied to study the structure of the upper mantle beneath the south-western margin of the East European Craton. The data were gathered in the passive seismic experiment "13 BB Star" (2013–2016) in the area of the crust recognized from previous borehole and refraction surveys. Several fundamental issues inherent in the linearised inversion were addressed in this work, including

- exploitation of a priori knowledge, choice of model's depth, trapping by local minima associated with non-uniqueness of the misfit-function optimization problem, proper weighting of data sets characterized by different uncertainties, and credibility of the final models. The last was investigated with novel 1D checkerboard tests juxtaposed with resolution matrix analysis. We advocate the usefulness of linearised approach when handled with proper care, and show that the resolution analysis is an indispensable step when choosing the inversion parameters. It allowed us to obtain reliable S-wave velocity models down
- to 200 km depth beneath the "13 BB Star" array, indicating the presence of a Paleozoic asthenosphere and the ceiling of the deeper, Precambrian, lithosphere-asthenosphere transition zone.

#### 1 Introduction

Limited resolvable depth and sensitivity are the two major problems inherent in receiver function (RF) and surface wave dispersion (SWD) analysis. The first issue results mainly from the presence of noise and imperfection of the measurements. The second constitutes the intrinsic feature of the data. Combined, they make the credible models of subsurface difficult to obtain. Although it is the inversion of RF which is usually regarded as exceptionally non-unique (Ammon and Randall, 1990;

Bodin et al., 2014), the SWD lacks uniqueness either, not being able to discriminate between the fine structures, especially at greater depths (Tsuboi and Saito, 1983; Romanowicz, 2002).

The simultaneous inversion of RF and SWD, having been applied to study deep lithosphere for 20 years (Özalaybey et al.,

1997; Du and Foulger, 1999; Julià et al., 2000), mitigates these issues thanks to the complementary sensitivity of the data (Shen et al., 2013). Even in this case however, there remains a certain ambiguity, showed in the following sections, that should be taken into careful consideration throughout inversion. A single model obtained by minimization of the data-misfit function cannot be regarded as a full solution of the inverse problem (Gubbins, 2004), which is often a case of linearised inversion (Julià et al., 2003; Horspool et al., 2006; Wang et al., 2014; Sosa et al., 2014; Bao et al., 2015; Li et al., 2016). One needs to probe

the whole space of physically plausible solutions, by drawing inferences from an ensemble (Sambridge and Mosegaard, 2002). Bayesian Monte Carlo methods (Green and Hastie, 2009; Bodin et al., 2012; Shen et al., 2013; Deng et al., 2015; Fontaine et al., 2015) appear to be a natural approach to achieve this goal, not only performing importance sampling (Sambridge, 1999), but also properly exploiting a priori knowledge of the problem (Malinverno, 2002). They are not flawless though, but suffer from their own inherent nuisance, e.g. the lack of objective criterion of convergence (Roberts et al., 1996, 1997), not to mention

5 their own inherent nuisance, e.g. the lack of objective criterion of convergence (Roberts et al., 1996, 1997), not to mention the computational cost. They also require nontrivial tuning to prevent solutions from following the shape of prior velocity distribution other than homogeneous (Minato et al., 2008; Wathelet, 2008).

Here we propose the compromise between the simplicity and transparency of the method on the one hand, and a full solution of the inverse problem on the other, by performing linearised inversion with ensemble of starting models covering entire space

10 of acceptable solutions. We demonstrate this approach with teleseismic data collected in the area of well-recognized crust – the a priori knowledge which we introduce into inversion. We show a remnant non-uniqueness of the joint RF and SWD inversion, which justifies the careful selection of the values of initial parameters. To this end, we perform the resolution analysis based on novel 1D checkerboard tests and resolution matrices before the actual inversion.

## 2 Data

25

Passive seismic experiment "13 BB Star" was dedicated to study the deep structure of the Earth's interior in the marginal zone of the East European Craton (EEC) in northern Poland (Grad et al., 2015). The seismic network consisted of 13 broadband stations on the area of ca. 120 km in diameter (Fig. 1). The network was located in the area of well-known sedimentary cover and crustal structure (Grad et al., 2009; Polkowski and Grad, 2015; Grad et al., 2016). The "13 BB Star" seismic stations were equipped with Reftek 151-120 Observer seismometers and Reftek 130 data logger. The stations were operated from June 2013
to October 2016. The distance and azimuthal epicentral distribution of analysed earthquakes are shown in Fig. 2.

#### 2.1 P receiver function

The receiver functions techniques (Langston, 1977; Vinnik, 1977) have been used to investigate the structure of the lithosphereasthenosphere system. Receiver functions were calculated by slightly modified method presented by Wilde-Piórko (2015); Wilde-Piórko et al. (2017). Seismograms used in receiver function analysis has been selected manually. Additionally, the second manual selection was done after the calculation of receiver functions to choose the traces with the highest signal to noise ratio. Ultimately, the total number of 99 events was taken into consideration. Seismograms were cut 300 s before and

- 300 s after the theoretical P-onset calculated for the *iasp91* model (Kennett and Engdahl, 1991) for earthquakes with epicentral distance of 30-100° and viewed to leave for further analysis only that ones with visible energy on vertical components. One-pass low-pass filtering with Butterworth filter of corner frequency 5 Hz was applied before resampling seismograms to 20 Hz.
- 30 Then, seismograms were cut in time window, 100 s before and 100 s after the onset of direct P-wave calculated due to *iasp91* model.

**Figure 1.** Location of "13 BB Star" broadband seismic stations on the background of tectonic map of north Poland and scheme of Europe. TESZ – Trans-European Suture Zone, Pre-Karelian granitoid massifs: Pm, Pomorze; Db, Dobrzyń; Cn, Ciechanów; RG, rapakivi granites; Pre-Karelian metamorphic belts: Kb, Kaszuby; Karelian metamorphic-magmatic complexes: Kmr (Kampinos); (Ryka, 1982).

Calculation of backazimuth and polarization angle were performed in two steps: (1) seismograms were filtered with two-pass band-pass Butterworth filter with corner periods of 2 and 10 s for searching the backazimuth angle for teleseismic P-wave; N and E components of seismograms were rotated with backazimuth angle from 0 to 360° every 1° and radial receiver function (RFR) were calculated by time-domain Wiener deconvolution; next RFR with maximal sum of amplitudes between time 0 and 1 s were selected (equivalent of rotation ZN E components to ZRT); (2) seismograms were filtered with two-pass band-pass Butterworth filter with corner periods of 2 and 10 s for searching the polarization angle for teleseismic P-wave; seismograms were rotated from ZNE to ZRT components by backazimuth angle found in step (1) and then rotated from ZRT to LQT with polarization angle from 0 to 45° every 1° and Q receiver function (RFQ) were calculated; next RFQ were cut 5 s before and 5 s after direct P-wave, a mean value and trend were removed and root square mean (rms) values were calculated for each RFQ

for time window between -2 and 0 s; then as best were chosen RFQ for which the rms value was lower than rms calculated for RFQ of next polarization angle (equivalent of rotation ZRT components to LQT). The final RF were filtered with Gaussian filter with parameter 4.

An example of presented procedure is shown in Fig. 3 together with RFs calculated in traditional way – ZNE components were rotated to LQT ones for theoretical backazimuth and polarization angles. The final RFR used in linearised joint inversion

were move-out corrected for mean slowness of each station and stacked (Fig. 4).

5