# Peer review of "Resolution analysis of joint inversion of seismic receiver function and surface wave dispersion curves in the "13 BB Star" experiment"

_Solid Earth, 2017_

## Referee Comment (RC1) · Anonymous Referee #1 · 4 Aug 2017

**General comments**

In their discussion paper, the authors present a joint inversion of Ps-receiver functions and Rayleigh wave dispersion for subsurface structure with special emphasis on the upper mantle, based on a newly collected dataset from the western margin of the East European Craton. The authors try to improve on the linearised inversion approach by incorporating detailed prior information on crustal structure, using a range of starting models, and optimising inversion parameters, e.g. weighting between data sets, damping, and range of data used, by synthetic tests and resolution analysis.

The new data set in this region of study, for which little broad-band seismological data

was available before, and a fresh approach to linearised inversion are of potential interest to the readership of Solid Earth. However, the results and the treatment of uncertainty are not convincing. The presented inversion results do not fit the measured data within their uncertainties (Figs. 11, 12), a problem that is not addressed at all, and different types of uncertainty, i.e. epistemic and aleatory, are mixed up. The observed Rayleigh wave dispersion curve shows anomalous dispersion (Fig. 12), with phase velocities decreasing from 100 to 180 s period, and the resulting velocity models show significantly higher S-wave velocities (5 km/s compared to 4.8 km/s or less) than results from other studies in cratonic areas, including Eastern Europe (Fig. 13). None of this is mentioned in the text, and though these might be novel and unusual observations, the explanation for the anomalous dispersion is very likely an unrecognised resolution issue in the surface wave analysis (array aperture of 120 km vs. wavelengths of 500 to 750 km for which measurements are shown). This requires a careful re-examination.

One of the main results mentioned in the abstract and conclusions is a velocity decrease around 180-200 km depth, which is at the edge of the depth resolution estimated by the authors. When considering the results for all stations as presented in Fig. 13 b), there are just four models out of 13 that show a clear decrease in velocity at their deeper end, though, and the average model (Fig. 13 a)) keeps nearly constant velocities beneath 170 km. It is not readily apparent how conclusions regarding the Precambrian lithosphere-asthenosphere transition (p. 1, l.10ff) can be drawn from these results, the more so as the inverted curves do not fit the measured data, the receiver functions do not seem to contain any significant information on structure at these depths (see below), and the measured dispersion curve is questionable at long periods.

The large misfit in inverting the receiver functions (Fig. 11) mainly concerns phases related to shallow structure (0-3s) and the Moho ($\sim 5.5$ s for the direct conversion and 18 s for the first multiple), where amplitudes of the inverted traces lie outside of the uncertainty of the measurements for both the inversion based on receiver functions only and the joint inversion. This is surely related to the a priori information on crustal

structure, which is "frozen" in the inversion, and, in case of the multiple, also to the fixed $v_P/v_S$ ratios and the move-out correction that is applied (and is only correct for direct phases). Crustal structure is not the main focus of this study. However, the a priori information clearly cannot explain the data, and as the phases that are not fit within their uncertainties are the ones with the largest amplitudes, the discrepancy in fitting them has a dominant influence on the misfit value for the receiver function data set. Actually, for the receiver function shown in Fig. 11, a zero line could fit the data within their uncertainty for all times after 7.5 s, with the exception of the crustal multiple. If all later amplitudes are not significantly different from zero, it is unclear how these data can help to constrain structure below 80-90 km depth. The only phase with an amplitude significantly different from zero that is likely related to the upper mantle, the negative wiggle around 7 s, is not fit well in the joint inversion example (Fig. 11 a) ).

Concerning the surface wave dispersion curve, it is not clear which standard deviation is used in the inversion, although the authors repeatedly emphasise that weighting by the uncertainties plays an important role (p. 7, l. 6 ff; p.9, l. 7ff). When showing the inversion results, no uncertainty is depicted for the dispersion curve, though (Fig. 12). The curve appears to be the same as shown in Fig. 5, but drawn to longer periods, which averages over the structure below the array as it is averaged over the tomographic maps that results from applying the ASWSM code (Jin and Gaherty, 2015). The $\pm 3\sigma$ deviation shown here is not the uncertainty in the measurements, but an expression of the variability in crustal structure below the array, which is not a random error. This also explains why errors decrease with period, i.e. depth, here, which is unphysical as resolution will decrease for larger wavelengths. The structure at larger depth will show less variability than the crust, however (which is also apparent from the prior information, Fig. 6). But the structural and geologic variability across the array, which is a type of epistemic uncertainty (e.g. Fig. 4 for the receiver functions), should not be confused with the measurement uncertainty, which is assumed to be random and thus aleatory. The measurement uncertainty of the dispersion curve could be estimated by using specific curves at each station that correspond to the cell in the phase velocity maps containing that station, and looking at the difference between maps for different events as the method of Jin and Gaherty (2015) averages phase velocity maps for individual events to arrive at the final map. Furthermore, if the kinks in the curve between 60 and 140 s are thought not to present real structure and can or should not be fit in the inversion, this has to be explained in the text, or the uncertainty at these periods should be large. Otherwise, it is not clear why the curve at periods below 60 s, which are important for upper mantle structure, is not fit by the inversion (Fig. 12).

My further concerns are explained in more detail below sequentially, following the structure of the text.

**Specific comments**
*Introduction*
I cannot agree with the first sentence in the introduction (p.1, l. 13) stating that limited resolvable depth is a major problem in receiver function and surface wave inversion. Converted phases from mantle transition zone discontinuities at 410 and 660 km depth are routinely analysed in Ps-receiver functions, and Rayleigh wave dispersion analysis, especially when including higher modes, likewise offers resolution to that depth (e.g. Legendre et al. 2012).

In the following, you compare the non-uniqueness in receiver function inversion to that inherent to the inversion of surface wave dispersion. However, these are two fundamentally different types of uncertainty. As receiver functions are a travel-time based method, there is a trade-off between the depth of a discontinuity and the velocity above in inversions (e.g. Ammon et al. 1990), and it is impossible to resolve both independently without additional constraints, at least when only a single ray parameter is considered. Surface waves do not suffer from a similar trade-off. Rather, they can constrain absolute velocities (e.g. Jin and Gaherty, 2015), but due to the long wavelengths considered, they average velocities over broader depth ranges (e.g. Julia et al. 2004, Fig. 4), thus the uncertainty is one in resolving power. Surface waves are not able to

resolve velocities in the individual thin layers used in the model parametrization here, but likewise, receiver functions would not be sensitive to velocity changes in individual 10m layers. Both of these uncertainties would also be present even if the data were perfect and noiseless, and are not just things usually assumed (p. 1, l. 16).

Some overview, which could be short, over the tectonic setting of the experiment is missing. Basically, Fig. 1 is all the information the reader gets, and for readers not familiar with the area, it is hard to understand the meaning and significance of the different features considered here. For example, various geological units beneath different array stations are not mentioned again, but the TESZ and the EEC are, and they are important for the expected structure in the upper mantle.

*Data: P receiver functions*
You mention "the receiver function techniques" (p. 2, l. 22). Probably, this should be "receiver function technique", as in the following, you only make references to Ps-receiver functions. Contrary to what is stated here, Ps-receiver functions alone have rarely been used to investigate the lithosphere-asthenosphere system, though - with some notable exceptions (Rychert and Shearer 2009), - due to the interference of upper mantle phases with crustal multiples. Widespread mapping of the lithosphere-asthenosphere boundary only started with the introduction of the Sp-receiver function technique (e.g. Vinnik and Farra 2000, Kind et al. 2012), which is only mentioned briefly in the discussion. Some further explanation would be helpful, if you actually want to refer to both "receiver function techniques".

The magnitude range of the earthquakes selected for receiver function calculation is not mentioned, but would be useful to know (i.e. minimum magnitude used in event selection).

The rotation of receiver functions into the LQT system based on the technique recently proposed by Wilde-Piorko et al. (2017) is described in detail (p. 3, ll. 1-12). The receiver functions used in the inversion are in the ZRT system, though, so the description of the second step consisting of rotation around the polarization angle to get from ZR to LQ is superfluous here.

The receiver functions are stacked after move-out correction (p. 3, ll. 15). The move-out correction can only be performed either for the direct conversions or for the multiples, though, meaning that multiples are degraded in amplitude and distorted in shape by stacking here, which does not allow for a meaningful fit during the inversion. If the data set is large enough, it would make more sense to stack traces over several limited distance or slowness ranges and use them as separate inputs to the inversion (e.g. Özaleybey et al. 1997, Julia et al. 2005, Dugda et al. 2007).

*Date: Rayleigh wave dispersion curves*
Were there any constraints regarding magnitude or hypocentral depth when selecting the events for surface wave processing?
Jin and Gaherty (2015) use their array method to recover phase velocities to 100 s period using USArray stations within 200 km distance of each other. Your array has an aperture of about 120 km, yet you measure phase velocities to 300 s (p. 6, l. 1). This would mean the distance across the array is about 1/10 of the wavelength, and considerably smaller for stations within the array. I severely doubt that phase velocities can still be reliably determined under these circumstances.
The recovered anomalous disperion at long periods is rather unusual in general, but also for a craton (e.g. Lebedev et al. 2009), with phase velocities decreasing by more than 0.1 km/s for periods between 100 and 180 s. Is this a resolution issue, or what is your interpretation otherwise?
The method of Jin and Gaherty (2015) imposes some smoothing constraints on the resulting phase velocity maps. How this might influence the resulting dispersion curves is not discussed here. The grid size used for the maps is likewise not given.
Due to the smoothing, the individual data points in the phase velocity maps are not independent. Additionally, the geological variation between different areas covered by the array (about 5 km variability in both sediment thickness and Moho depth according to the prior data shown in Fig. 6a) ) is likely not random. Thus, it makes no sense to average over the array and treat the resulting standard deviation as random error in the inversion.

*Inversion*
The meaning of $\Delta r_i$ and $\Delta s_i$ in formula (1) is not explained (difference between observed and modelled data), and it should be stated somewhere that the aim of the inversion is to (iteratively) minimize the misfit function.
Besides, the model parametrisation is not explained - what is contained in the parameter vector m which is inverted for (S-wave velocities)?
There is something wrong in p. 7, l. 22. The values listed here look like $v_P/v_S$ ratios, but they have units of km/s.
While an average curve is used for Rayleigh wave dispersion, receiver functions are inverted separately for each station. Is this warranted for the upper mantle by the inter-station distance, or do sampled regions overlap? How large is the area covered by piercing points for an individual station at, for example, 150 km depth and what is the Fresnel zone width?
The rational for including the prior information is that it is a good and detailed representation of crustal structure in the area, so that the inversion results for the crust should not differ too much from it. However, the prior data cannot explain the crustal part (first 6-7 s) of the observed receiver functions, even when allowing for some variation during the inversion. This issue is not addressed at all, though it is the major reason for a large misfit to the observed receiver functions in the inversion, which casts some doubt on the usefulness of including the prior information in this particular way.
The paragraph on tuning the inversion parameters by synthetic tests is too brief. It does not give any idea on how many tests with which kind of models were conducted and how they were evaluated, e.g. were trade-off curves between resolution and stability like in Julia et al. (2000) used? If the tests referred to are the checkerboard tests explained in more detail in the following paragraph, that should be clearly stated here.

*Resolution analysis*
It is not clear to me why the checkerboard resolution test from tomography - which is actually also used in 3D, and not just in 2D as stated here - should be applicable to 1D joint inversion of receiver functions and surface wave dispersion. In tomography,

resolution is determined by the density and directional variability of rays in each grid cell of the covered region, and the input to the checker-board test are travel-times that average over the region with the anomaly pattern. Here, the input are the receiver function waveforms, which will show additional phases for each impedance contrast added to the model by the checkerboard anomalies in a rather non-linear way, if these anomalies are large enough, and a dispersion curve that will average the anomalies over some depth intervals. The ability to resolve these structures does not depend on data coverage, but on how well they are separated and how large their amplitudes are, as well as on their interaction with the background model (at least this seems to be indicated by the dependence on polarity). In tomography, the checkerboard test helps to select the size of grid cells in the model and the value of a damping parameter to obtain optimal resolution as well as to decide which areas are not well resolved. Here, you try to optimize various inversion parameters via the test, but it is not clear if there are trade-offs between parameters selected independently (for example, could a higher value of $\theta$ reduce the amplitude of the oscillations in Fig. 8c)? And might the inability to fit higher amplitude anomalies in Fig. 7d) be due to the choice of the damping parameter?) The size and distribution of the anomaly pattern also has an influence on how well it can be recovered (p. 10, l. 2ff), which is not comparable to observations from tomography. It is not clear to me why the polarity of the input pattern should influence the recovery of the anomalies and what this means for the inversion results. Anomalies can only be resolved in some depth if they are positive, but not if they are negative? What does this mean for the derived velocity models? That anomalies might be missed depending on their polarity and depth?

What is the difference between Fig. 7 b), with selected model parameters, and Fig. 8 a), with parameters described as poorly tuned? To me, they look about the same, i.e. positive anomalies are better resolved than negative ones, and below 140 km, resolution is generally bad and the inverted anomalies do not compare well with the input pattern.

The description of Fig. 7 d) is unclear - what is meant by "5% of the anomaly pattern"?
The anomaly size is given in km/s, and it's not clear what the 5% refer to. The amplitude of the anomalies in the other tests (Fig. 7 a)–c)) should also be mentioned. And what is the criterion for the vertical size of the anomaly patterns chosen, each anomaly covering 5 layers?

In formula (5), the meaning of matrices **A** and $\mathbf{C}_e$ and $\mathbf{C}_m$ is not explained.

*Results*

Some information is missing to interpret the rainbow-coloured set of curves in Fig. 10, 11 and 12. Do they refer to different starting models for the mantle with increasing velocity from blue to red, as depicted in Fig. 6b)? Or does blue to red mean increasing misfit of acceptable models, and in what range? It seems that the misfit for the red curves is largest, but why are they plotted on top then? This makes them visually the most striking elements of the curve ensemble and gave me at first the impression that they must be the best-fitting curves. Specifically with regard to the velocity models in Fig. 10, some information on which models are the best fitting is needed, as velocity decreases between 150 and 200 km mainly appear in the models plotted in red. How do you determine which models are acceptable and which not, given that all inverted receiver functions and dispersion curves lie outside the uncertainty of the observed data?

The obtained S-wave velocity models are rather fast, even for cratons, compared to e.g. Fischer et al. (2010), Lebedev et al. (2009), Meier et al. (2016) and Vinnik et al. (2015), where velocities in the depth range considered here are generally below 4.8 km/s. The extraordinary fast velocities obtained here (5 km/s or more for all stations below about 150 km, Fig. 13b)) need some further discussion and explanation.

*Inadequate and missing references*

The reference to Wathelet et al. (2008) on p. 2, l. 7 is in a wrong context. The improved Neighbourhood Algorithm as described in that study is a direct sampling technique, where constraints are only imposed as boundaries on the searchable parameter space, not in the form of a priori probability distributions, and the study does not deal with

Bayesian inference based on the NA results, as, e.g. Sambridge (1999). Accordingly, Wathelet et al. (2008) does not discuss the influence of prior velocity distributions.

The reference to Jones et al. (2010) on p. 8, l. 6 is in a wrong context. Jones et al. discuss estimates for the depth of the lithosphere-asthenosphere boundary in central Europe, including one based on a P-wave tomography. They do not mention any S-wave velocities, though. Likewise, Kind et al. (2015) present migrated receiver function and do not discuss absolute values for mantle velocities beneath cratons.

References to other linearised inversion attempts that used a number of starting models covering the complete likely parameter space, similar to what is proposed here, are missing. For example, Graw et al. (2017) use 24 smoothed and perturbed versions of the ak135 model as starting models in the inversion of P-wave responses in Antarctica.

Reference to the recent PASSEQ project and upper mantle studies based on these data using surface waves (Meier et al. 2016, Soomro et al. 2016), Ps- and Sp-receiver functions (Knapmeyer-Endrun et al. 2017), or trying to combine Sp-receiver functions and surface waves (Kind et al. 2017) are absent; only the body wave tomography based on PASSEQ data is referred to (Janutyte et al. 2015), though likewise not in the context of its results for upper mantle structure at the edge of the East European craton. Though the "13 BB Star" experiment used here surely has a better resolution and data quality in the region of study, the PASSEQ network also contained temporary stations in this area and results obtained from these data should at least be mentioned in the discussion, especially if they use the same methods of analysis. The discussion should also give some idea why the Rayleigh wave dispersion curves displayed in Meier et al. (2016), which show increasing phase velocities for periods above 100 s, are so different from the one derived here, which shows decreasing velocities above 100 s, whereas the inverted velocity models contain significantly higher S-wave velocities of 5 km/s (vs. 4.7 km/s for the craton model in Meier et al.) below 150 km.

**Technical corrections**

There are some typings and grammar issues, i.e.

p. 1, l. 23: "a case of linearised inversion" should be "the case for linearised inversion"

p. 2, l. 24: "has" should be "have"

p. 2., l. 28: "that" should be "the"

p. 4, l. 5 and l. 12: "form" should be "from"

p. 4, l. 11ff: "In average" should be "On average"

p. 7, l. 16: "to" is doubled ("analogy to to Bayesian inference")

p. 7, l. 17: brackets around "freeze" should be removed

p. 7, l. 21: brackets around references to Berteussen (1977) and Gardner et al. (1974) should be removed

p. 9, l. 1: "fit into" should be "fit the"

p. 16, l. 1ff: the reference to Ammon et al. (1990) misses one of the three authors (G. Zandt), and there's a typing error in the page number (should be 15318 instead of 153318)

p. 19, l. 7ff: "Schulte-Pelkum", and a missing space between "Bayesian" and "Monte Carlo"

Quite a few articles, both definite and indefinite, are missing, e.g. on p. 4, ll. 5-9. It might be useful to have a native speaker check the text in this respect.

**References**

Ammon et al. (1990), JGR, 95, 15303-15318

Fischer et al. (2010), Ann. Rev. Earth Planet. Sci., 38, 551-575

Graw et al. (2017), BSSA, 107, 63-651, doi:10.1785/0120160262

Dugda et al. (2007), JGR, 112, 2156-2202

Janutyte et al. (2015), Solid Earth, 6, 73-91

Jin and Gaherty (2015), GJI, 201, 1383-1398, doi:10.1093/gji/ggv079

Jones et al. (2010), Lithos, 120, 14-29, doi:10.1016/j.lithos.2010.07.013

Juliá et al. (2000), GJI, 143, 95-112

Juliá et al. (2005), GJI, 162, 555-569, doi:10.1111/j.1365-246X.2005.02685.x
Kind et al. (2012), Tectonophysics, 536–537, 25–43, doi:10.1016/j.tecto.2012.03.005

Kind et al. (2015), Solid Earth, 6, 975-970, doi:10.5194/se-6-957-2015

Kind et al. (2017), Tectonophysics, 700-701, 19-31, doi:10.1016/j.tecto.2017.02.002

Knapmeyer-Endrun et al. (2017), EPSL, 458, 429-441, doi:10.1016/j.epsl.2016.11.011

Lebedev et al. (2009), Lithos, 109, 96-111, doi:10.1016/j.lithos.2008.06.010

Legendre et al. (2012), GJI, 191, 282-304, doi:10.1111/j.1365-246X.2012.05613.x

Meier et al. (2016), Tectonophysiscs, 692, 58-73, doi:10.1016/j.tecto.2016.09.016

Özalaybey et al. (1997), BSSA, 87, 183-199

Rychert and Shearer (2009), Science, 324, 495-498, doi:10.1126/science.1169754

Soomro et al. (2016), GJI, 204, 517-534, doi:10.1093/gji/ggv462

Vinnik and Farra (2000), GJI, 141, 699-712

Vinnik et al. (2015), Tectonophysics, 667, 189-198

Wathelet (2008), GRL, 35, L09301, doi:10.1029/2008GL033256

---

## Referee Comment (RC2) · Anonymous Referee #2 · 16 Aug 2017

The manuscript is about the joint inversion of Rayleigh wave phase velocity dispersion and P receiver function applied to 13 broadband stations that have been recording for 3 years at the south-western margin of the East European Craton. The manuscript wants to give some new input on the application of the linearized inversion, and to give constraints on the mantle structure of the study area. In my opinion this paper is too synthetic; fundamental sections, like results and discussions are superficially written, while they need a longer, accurate and also descriptive argumentation, in order to demonstrate the quality and meaning of the results. In its actual shape the paper is poor and raw, and misses the accurate descriptions needed to deserve publication.

[Figure]

In the following I am listing specific problems of the manuscript.

1) The results of the RF inversion only are better (although not yet fully convincing, see next point) than the results of the joint inversion of RF and SWD (Figure 11). Therefore it is not clear why the authors spend their time applying the joint inversion when inversion the RF only could give better results. If the authors want to proof that the joint inversion gives better constraints for unraveling the subsurface structure, then they have to convince the reader by adding examples (and explaining them), and with some argumentation that is lacking at the moment.

2) The chosen crustal model (and "frozen") for the inversion is clearly not correct for the area. The fit between observed and synthetic RF for the initial 5 s is poor. If the paper wants to address the issue of "exploiting a priori knowledge" (as stated in the abstract), then the authors should show what happens in the inversion if the shallow part of the model is free and not "frozen", and show how their results are improved.

3) In the same way in the text it is not explained how the joint inversion improves (or not improves) the results of the SWD inversion only.

4) The description of the results is almost lacking, it actually consists in listing the number of figures that show the results, such figures are not well described as well, and their meaning and their importance is never mentioned as well.

5) The discussion section is extremely short and it does not add anything new to previous knowledge, probably because the paper has nothing new to add to the state of the art of both the technique and structural features of the area. The following "promises" made in the abstract: "Several fundamental issues inherent in the linearised inversion were addressed in this work, including exploitation of a priori knowledge, choice of model's depth, trapping by local minima associated with non-uniqueness of the misfit-function optimization problem, proper weighting of data sets characterized by different uncertainties, and credibility of the final models" must be explained and discussed in this section.

6) Figure4: the several RF stacks plotted on top of each other are hardly comprehensible. Each stack must be plotted singularly, for seek of clarity.

7) Acronyms such as ASWMS, CPS, FWI must be explicated somewhere in the text

8) Figures 10,11, and 12 deserve a complete caption; the colors in these figures are not explained at all

Technical corrections:

Page 1 line 4: linearised → linearized

Page 2, line 9: covering THE entire

Page 2 lines 26-28: the sentence is badly written, and should be rewritten

Page 8 Line 3: cover → covers

---

## Author Comment (AC1) · 5 Dec 2017

Interactive comment on "Resolution analysis of joint inversion of seismic receiver function and surface wave dispersion curves in the "13 BB Star" experiment" by Kajetan Chrapkiewicz et al. Anonymous Referee #1 General comments In their discussion paper, the authors present a joint inversion of Ps-receiver functions and Rayleigh wave dispersion for subsurface structure with special emphasis on the upper mantle, based on a newly collected dataset from the western margin of the East European Craton. The authors try to improve on the linearised inversion approach by incorporating detailed prior information on crustal structure, using a

range of starting models, and optimising inversion parameters, e.g. weighting between data sets, damping, and range of data used, by synthetic tests and resolution analysis. The new data set in this region of study, for which little broad-band seismological data was available before, and a fresh approach to linearised inversion are of potential interest to the readership of Solid Earth. However, the results and the treatment of uncertainty are not convincing. The presented inversion results do not fit the measured data within their uncertainties (Figs. 11, 12), a problem that is not addressed at all, and different types of uncertainty, i.e. epistemic and aleatory, are mixed up.

1. Revisiting our inversion workflow allowed us to significantly improve the fit for both data sets (see Fig. 14 and 17 of the revised manuscript). The uncertainty of the SWD has also been reassessed with the method similar to the one suggested by the reviewer and described in Subs. 2.2 and now error bands increase with period (Fig. 4b of the revised manuscript). Figure suppl-fig01 attached to this reply shows the cloud of points (each point represents one cell of the map averaged over all events for a given period) taken for the final average. As has been pointed out, averaging over the map cells introduces to the estimation an additional, epistemic component representing the geologic variation of the structure beneath the array yet this can be neglected except for the shortest periods. We use those, however, effectively only for a crustal inversion, constrained with RF data, which wasn't whatsoever the main target of this study (see Sect. 5 of the revised manuscript).

The observed Rayleigh wave dispersion curve shows anomalous dispersion (Fig. 14), with phase velocities decreasing from 100 to 180 s period, and the resulting velocity models show significantly higher S-wave velocities (5 km/s compared to 4.8 km/s or less) than results from other studies in cratonic areas, including Eastern Europe (Fig. 15). None of this is mentioned in the text, and though these might be novel and unusual observations, the explanation for the anomalous dispersion is very likely an unrecognised resolution issue in the surface wave analysis (array aperture of 120 km vs. wavelengths of 500 to 750 km for which measurements are shown). This requires

a careful reexamination.

2. After reexamination of our data constraining the mantle (see Sect. 5 of revised manuscript: "The maximum period of SWD was extended to 180 s, but the time window of RF remained the same. The true amplitudes of reflected phases had been lost due to the move-out correction and stacking, and the time series after direct Moho conversion wouldn't be reliable to invert. Because our RF data set wasn't large enough, we couldn't stack RFs over different slowness ranges and use them as separate inputs in the inversion. For this reason, RF didn't constrained most of the mantle and only smooth mantle structures were expected in the second-stage inversion."), the method we used to calculate the SWD curve, as well as the depth of the model used in inversion (900 km vs previous 300 km), we obtained distinctively slower and smoother final models (see Fig. 14b of the revised manuscript). As regards the resolution, the distinct phase shifts between stations were visible for some events even for periods as high as 250 s. Please see attached figures with filtered seismograms for aĂăsingle event 2014-04-01 (M8.2) and stations A0, B2 and B5 lying on its great circle (deviation less than 13 deg): - suppl-fig12: A0 station in red, B5 station in black, - suppl-fig13: B2 station in red, B5 station in black.

One of the main results mentioned in the abstract and conclusions is a velocity decrease around 180-200 km depth, which is at the edge of the depth resolution estimated by the authors. When considering the results for all stations as presented in Fig. 15 b), there are just four models out of 13 that show a clear decrease in velocity at their deeper end, though, and the average model (Fig. 15 a)) keeps nearly constant velocities beneath 170 km. It is not readily apparent how conclusions regarding the Precambrian lithosphere-asthenosphere transition (p. 1, l.10ff) can be drawn from these results, the more so as the inverted curves do not fit the measured data, the receiver functions do not seem to contain any significant information on structure at these depths (see below), and the measured dispersion curve is questionable at long periods.

[Figure]

3. After the revision all the models exhibit similar feature at these depths (Fig. 14b), the fit also improved (Fig. 15). The mantle part of the results was modeled based on the mean SWD curve for the whole array with no contribution of RF data any more.

The large misfit in inverting the receiver functions (Fig. 13) mainly concerns phases related to shallow structure (0-3s) and the Moho (âĹij 5.5 s for the direct conversion and 18 s for the first multiple), where amplitudes of the inverted traces lie outside of the uncertainty of the measurements for both the inversion based on receiver functions only and the joint inversion. This is surely related to the a priori information on crustal structure, which is "frozen" in the inversion, and, in case of the multiple, also to the fixed vP /vS ratios and the move-out correction that is applied (and is only correct for direct phases). Crustal structure is not the main focus of this study. However, the a priori information clearly cannot explain the data, and as the phases that are not fit within their uncertainties are the ones with the largest amplitudes, the discrepancy in fitting them has a dominant influence on the misfit value for the receiver function data set. Actually, for the receiver function shown in Fig. 13, a zero line could fit the data within their uncertainty for all times after 7.5 s, with the exception of the crustal multiple. If all later amplitudes are not significantly different from zero, it is unclear how these data can help to constrain structure below 80-90 km depth. The only phase with an amplitude significantly different from zero that is likely related to the upper mantle, the negative wiggle around 7 s, is not fit well in the joint inversion example (Fig. 13 a) ).

4. We agree with these statements. Please see point 2 of this answer.

Concerning the surface wave dispersion curve, it is not clear which standard deviation is used in the inversion, although the authors repeatedly emphasise that weighting by the uncertainties plays an important role (p. 7, l. 6 ff; p.9, l. 7ff).

5. The uncertainty used in the inversion is shown in Fig. 4b of the revised manuscript. This weighting plays a role within SWD data vector indeed, but the whole vector is

weighed also with respect to RF vector by influence parameter p.

When showing the inversion results, no uncertainty is depicted for the dispersion curve, though (Fig. 14). The curve appears to be the same as shown in Fig. 5, but drawn to longer periods, which averages over the structure below the array as it is averaged over the tomo- graphic maps that results from applying the ASWSM code (Jin and Gaherty, 2015). The $\pm\,3\sigma$ deviation shown here is not the uncertainty in the measurements, but an expression of the variability in crustal structure below the array, which is not a random error. This also explains why errors decrease with period, i.e. depth, here, which is unphysical as resolution will decrease for larger wavelengths. The structure at larger depth will show less variability than the crust, however (which is also apparent from the prior information, Fig. 6). But the structural and geologic variability across the array, which is a type of epistemic uncertainty (e.g. Fig. 4 for the receiver functions), should not be confused with the measurement uncertainty, which is assumed to be random and thus aleatory. The measurement uncertainty of the dispersion curve could be estimated by using specific curves at each station that correspond to the cell in the phase velocity maps containing that station, and looking at the difference between maps for different events as the method of Jin and Gaherty (2015) averages phase velocity maps for individual events to arrive at the final map.

6. Please see Fig. 4b, 11b, 13b of the revised manuscript and point 1 of this response.

Furthermore, if the kinks in the curve between 60 and 140 s are thought not to present real structure and can or should not be fit in the inversion, this has to be explained in the text, or the uncertainty at these periods should be large. Otherwise, it is not clear why the curve at periods below 60 s, which are important for upper mantle structure, is not fit by the inversion (Fig. 14).

7. Please see Sect. 6 of the revised manuscript (" The inability of the synthetic curves to fit the notch present in the observed SWD curve between $60-90$ s is likely related to the oversimplified forward calculations assuming e.g. isotropy of the model.") We observed similar feature on group velocity curves as well, and claim that it may represent the real, probably anisotropic structure beyond modeling capability of our code.

My further concerns are explained in more detail below sequentially, following the structure of the text. Specific comments Introduction I cannot agree with the first sentence in the introduction (p.1, l. 13) stating that limited resolvable depth is a major problem in receiver function and surface wave inversion. Converted phases from mantle transition zone discontinuities at 410 and 660 km depth are routinely analysed in Ps-receiver functions, and Rayleigh wave dispersion analysis, especially when including higher modes, likewise offers resolution to that depth (e.g. Legendre et al. 2012). In the following, you compare the non-uniqueness in receiver function inversion to that inherent to the inversion of surface wave dispersion. However, these are two fundamentally different types of uncertainty. As receiver functions are a travel-time based method, there is a trade-off between the depth of a discontinuity and the velocity above in inversions (e.g. Ammon et al. 1990), and it is impossible to resolve both independently without additional constraints, at least when only a single ray parameter is considered. Surface waves do not suffer from a similar trade-off. Rather, they can con- strain absolute velocities (e.g. Jin and Gaherty, 2015), but due to the long wavelengths considered, they average velocities over broader depth ranges (e.g. Julia et al. 2004, Fig. 4), thus the uncertainty is one in resolving power. Surface waves are not able to resolve velocities in the individual thin layers used in the model parametrization here, but likewise, receiver functions would not be sensitive to velocity changes in individual 10m layers. Both of these uncertainties would also be present even if the data were perfect and noiseless, and are not just things usually assumed (p. 1, l. 16).

8. It was more badly written than meant what was understood here. Please see more concise and hopefully clear version of the introduction in the revised manuscript.

Some overview, which could be short, over the tectonic setting of the experiment is missing. Basically, Fig. 1 is all the information the reader gets, and for readers not familiar with the area, it is hard to understand the meaning and significance of the

different features considered here. For example, various geological units beneath different array stations are not mentioned again, but the TESZ and the EEC are, and they are important for the expected structure in the upper mantle.

9. Please see Fig. 01 and the beginning of Sect. 2 of the revised manuscript. We don't want to focus more on tectonic setting because the paper is meant to be rather technical.

Data: P receiver functions You mention "the receiver function techniques" (p. 2, l. 22). Probably, this should be "receiver function technique", as in the following, you only make references to Ps- receiver functions. Contrary to what is stated here, Ps-receiver functions alone have rarely been used to investigate the lithosphere-asthenosphere system, though - with some notable exceptions (Rychert and Shearer 2009), - due to the interference of upper mantle phases with crustal multiples. Widespread mapping of the lithosphere- asthenosphere boundary only started with the introduction of the Sp-receiver function technique (e.g. Vinnik and Farra 2000, Kind et al. 2012), which is only mentioned briefly in the discussion. Some further explanation would be helpful, if you actually want to refer to both "receiver function techniques".

10. Corrected.

The magnitude range of the earthquakes selected for receiver function calculation is not mentioned, but would be useful to know (i.e. minimum magnitude used in event selection).

11. The events were of the magnitude 5.7 and higher, we included this information in Subsect 2.1 of the revised manuscript.

The rotation of receiver functions into the LQT system based on the technique recently proposed by Wilde-Piorko et al. (2017) is described in detail (p. 3, ll. 1-12). The receiver functions used in the inversion are in the ZRT system, though, so the description of the second step consisting of rotation around the polarization angle to get from ZR

to LQ is superfluous here.

12. LQT part has been deleted.

The receiver functions are stacked after move-out correction (p. 3, ll. 15). The move-out correction can only be performed either for the direct conversions or for the multiples, though, meaning that multiples are degraded in amplitude and distorted in shape by stacking here, which does not allow for a meaningful fit during the inversion. If the data set is large enough, it would make more sense to stack traces over several limited distance or slowness ranges and use them as separate inputs to the inversion (e.g. Özaleybey et al. 1997, Julia et al. 2005, Dugda et al. 2007).

13. We agree, please see point 2 of this reply.

Date: Rayleigh wave dispersion curves Were there any constraints regarding magnitude or hypocentral depth when selecting the events for surface wave processing?

14. M>4, no depth constraints. We included this information in Subsect 2.2 of the revised manuscript.

Jin and Gaherty (2015) use their array method to recover phase velocities to 100 s period using USArray stations within 200 km distance of each other. Your array has an aperture of about 120 km, yet you measure phase velocities to 300 s (p. 6, l. 1). This would mean the distance across the array is about 1/10 of the wavelength, and considerably smaller for stations within the array. I severely doubt that phase velocities can still be reliably determined under these circumstances. The recovered anomalous disperion at long periods is rather unusual in general, but also for a craton (e.g. Lebedev et al. 2009), with phase velocities decreasing by more than 0.1 km/s for periods between 100 and 180 s. Is this a resolution issue, or what is your interpretation otherwise?

15. Please see point 2 of this reply.

The method of Jin and Gaherty (2015) imposes some smoothing constraints on the

resulting phase velocity maps. How this might influence the resulting dispersion curves is not discussed here.

16. No smoothing has been applied. We included this information in Subsect 2.2 of the revised manuscript.

The grid size used for the maps is likewise not given.

17. Grid size was 0.05 deg x 0.05 deg. We included this information in Subsect 2.2 of the revised manuscript.

 Due to the smoothing, the individual data points in the phase velocity maps are not independent. Additionally, the geological variation between different areas covered by the array (about 5 km variability in both sediment thickness and Moho depth according to the prior data shown in Fig. 6a) ) is likely not random. Thus, it makes no sense to average over the array and treat the resulting standard deviation as random error in the inversion.

18. Please see point 1 of this reply.

Inversion The meaning of $\Delta r_i$ and $\Delta s_i$ in formula (1) is not explained (difference between observed and modelled data),

19. Corrected.

and it should be stated somewhere that the aim of the inversion is to (iteratively) minimize the misfit function.

20. Corrected, please see Sect. 3 of the revised manuscript (The ultimate goal is to find all models explaining the data and remaining consistent with the prior knowledge of the problem. S-wave velocity was the only parameter inverted here. P-wave velocities were updated based).

Besides, the model parametrisation is not explained - what is contained in the parameter vector m which is inverted for (S-wave velocities)?

[Figure]

21. Corrected, please see Sect. 3 of the revised manuscript. (S-wave velocity was the only parameter inverted here.).

There is something wrong in p. 7, l. 22. The values listed here look like vP /vS ratios, but they have units of km/s. 

22. Typos corrected.

While an average curve is used for Rayleigh wave dispersion, receiver functions are inverted separately for each station. Is this warranted for the upper mantle by the inter-station distance, or do sampled regions overlap? How large is the area covered by piercing points for an individual station at, for example, 150 km depth and what is the Fresnel zone width?

23. This remark is not valid any more, because we have revisited our inversion workflow (see point 3 of this reply). The Fresnel zone width is more than 50 km at 50 km depth for all phases and periods considered in RF, whereas piercing points are 12 km and 40 km for discontinuities at 50 and 150 km depth respectively (see supplementary figures suppl-fig2-7).

The rational for including the prior information is that it is a good and detailed represen-tation of crustal structure in the area, so that the inversion results for the crust should not differ too much from it. However, the prior data cannot explain the crustal part (first 6-7 s) of the observed receiver functions, even when allowing for some variation during the inversion. This issue is not addressed at all, though it is the major reason for a large misfit to the observed receiver functions in the inversion, which casts some doubt on the usefulness of including the prior information in this particular way.

24. Please see point 1 of this reply.

The paragraph on tuning the inversion parameters by synthetic tests is too brief. It does not give any idea on how many tests with which kind of models were conducted and how they were evaluated, e.g. were trade-off curves between resolution and stability

like in Julia et al. (2000) used? If the tests referred to are the checkerboard tests explained in more detail in the following paragraph, that should be clearly stated here.

25. Please see Sect. 3.2 and first paragraph of Sect. 4.2.2 of the revised manuscript. The synthetic analysis used here replaced the trade-off curves, being their, maybe less rigorous, but more intuitive extension.

Resolution analysis It is not clear to me why the checkerboard resolution test from tomography - which is actually also used in 3D, and not just in 2D as stated here

26. Of course it's not limited to 2D, corrected.

- should be applicable to 1D joint inversion of receiver functions and surface wave dispersion. In tomography, resolution is determined by the density and directional vari- ability of rays in each grid cell of the covered region, and the input to the checker-board test are travel-times that average over the region with the anomaly pattern. Here, the input are the receiver function waveforms, which will show additional phases for each impedance contrast added to the model by the checkerboard anomalies in a rather non-linear way, if these anomalies are large enough, and a dispersion curve that will average the anomalies over some depth intervals. The ability to resolve these struc- tures does not depend on data coverage, but on how well they are separated and how large their amplitudes are, as well as on their interaction with the background model (at least this seems to be indicated by the dependence on polarity). In tomography, the checkerboard test helps to select the size of grid cells in the model and the value of a damping parameter to obtain optimal resolution as well as to decide which areas are not well resolved.

27. Our checkerboard test shares with tomography only its methodology, but the rea- soning behind it is indeed quite different and shouldn't be compared in any way. Sure enough, resolving power of joint inversion of RF and SWD depends on other factors, and this is what this test is meant to show among other things. A failure to recover the anomalies is indicative of no resolving power of the method regardless of the nature of

the wave phenomena behind it.

Here, you try to optimize various inversion parameters via the test, but it is not clear if there are trade-offs between parameters selected independently (for example, could a higher value of $\theta$ reduce the amplitude of the oscillations in Fig. 8c)?

28. It could have done since there is a cross-talk between these parameters indeed. Please see Sect. 3.2 (Note that higher values of p (favouring the dispersion data more) might smoothen the results similarly to the increase of $\theta 2$ . Therefore we checked the optimal value of $\theta 2$ through the inversions of aÂăsingle data type as well.) of the revised manuscript.

And might the inability to fit higher amplitude anomalies in Fig. 7d) be due to the choice of the damping parameter?)

29. Our tests showed that decreasing the damping in the case of stronger anomalies rather introduces artifacts than improves recovery.

The size and distribution of the anomaly pattern also has an influence on how well it can be recovered (p. 10, l. 2ff), which is not comparable to observations from tomography.

30. Again, these two shouldn't be compared. For more details see point 31 of this reply.

It is not clear to me why the polarity of the input pattern should influence the recovery of the anomalies and what this means for the inversion results. Anomalies can only be resolved in some depth if they are positive, but not if they are negative? What does this mean for the derived velocity models? That anomalies might be missed depending on their polarity and depth?

31. This phenomenon indicates asymmetry of the starting models relative to the true one, and possible inability to recover some of the anomalies, indeed. Please see Fig. 09 and Subsect. 4.2.1 of the revised manuscript for more detailed example.

What is the difference between Fig. 7 b), with selected model parameters, and Fig. 8 a), with parameters described as poorly tuned? To me, they look about the same, i.e. positive anomalies are better resolved than negative ones, and below 140 km, resolution is generally bad and the inverted anomalies do not compare well with the input pattern.

32. This is true. Here parameters are regarded as well tuned if they recover at least some of the patterns (e.g. positive anomaly like in this case). Poorly tuned parameters don't recover any.

The description of Fig. 7 d) is unclear - what is meant by "5% of the anomaly pattern"?

33. Background model. Corrected.

The anomaly size is given in km/s, and it's not clear what the 5% refer to.

34. Plots show the true amplitude in km/s since it's varies with the changes of the background model. Labeling with constant "5 % of background model" wouldn't be insightful in our opinion.

The amplitude of the anomalies in the other tests (Fig. 7 a)–c)) should also be mentioned.

35. Corrected.

And what is the criterion for the vertical size of the anomaly patterns chosen, each anomaly covering 5 layers?

36. Sizes varied in different cases to test different length scales. They were preferably multiples of a single-layer thickness, otherwise their size would vary with depth.

In formula (5), the meaning of matrices A and Ce and Cm is not explained.

37. Corrected.

Results Some information is missing to interpret the rainbow-coloured set of curves

in Fig. 12, 11 and 12. Do they refer to different starting models for the mantle with increasing velocity from blue to red, as depicted in Fig. 6b)?

38. Yes. Please see captions below Fig. 12-13 in the revised manuscript.

Or does blue to red mean increasing misfit of acceptable models, and in what range? It seems that the misfit for the red curves is largest, but why are they plotted on top then? This makes them visually the most striking elements of the curve ensemble and gave me at first the impression that they must be the best-fitting curves. Specifically with regard to the velocity models in Fig. 12, some information on which models are the best fitting is needed, as velocity decreases between 150 and 200 km mainly appear in the models plotted in red. How do you determine which models are acceptable and which not, given that all inverted receiver functions and dispersion curves lie outside the uncertainty of the observed data?

39. In the revised version we show results for all starting models, without plotting the average. They lie very close to each other except for the part just below the Moho (Fig. 14b of the revised manuscript) which accounts for the difference in RF (Fig. 15a of the revised manuscript). Rejecting dark-blue models with RFs falling outside acceptable fit range and plotting the mean model could be done but doesn't add much to this section in our opinion.

The obtained S-wave velocity models are rather fast, even for cratons, compared to e.g. Fischer et al. (2010), Lebedev et al. (2009), Meier et al. (2016) and Vinnik et al. (2015), where velocities in the depth range considered here are generally below 4.8 km/s. The extraordinary fast velocities obtained here (5 km/s or more for all stations below about 150 km, Fig. 15b)) need some further discussion and explanation.

40. Please see point 2 of this reply and the last paragraph of Sect. 6 of the revised manuscript.

Inadequate and missing references The reference to Wathelet et al. (2008) on p. 2, l.

7 is in a wrong context. The improved Neighbourhood Algorithm as described in that study is a direct sampling technique, where constraints are only imposed as boundaries on the searchable parameter space, not in the form of a priori probability distributions, and the study does not deal with Bayesian inference based on the NA results, as, e.g. Sambridge (1999). Accordingly, Wathelet et al. (2008) does not discuss the influence of prior velocity distributions.

41. Corrected.

The reference to Jones et al. (2010) on p. 8, l. 6 is in a wrong context. Jones et al. discuss estimates for the depth of the lithosphere-asthenosphere boundary in central Europe, including one based on a P-wave tomography. They do not mention any S-wave velocities, though. Likewise, Kind et al. (2015) present migrated receiver function and do not discuss absolute values for mantle velocities beneath cratons.

42. Corrected.

References to other linearised inversion attempts that used a number of starting models covering the complete likely parameter space, similar to what is proposed here, are missing. For example, Graw et al. (2017) use 24 smoothed and perturbed versions of the ak135 model as starting models in the inversion of P-wave responses in Antarctica.

43. Corrected.

Reference to the recent PASSEQ project and upper mantle studies based on these data using surface waves (Meier et al. 2016, Soomro et al. 2016), Ps- and Sp-receiver functions (Knapmeyer-Endrun et al. 2017), or trying to combine Sp-receiver functions and surface waves (Kind et al. 2017) are absent; only the body wave tomography based on PASSEQ data is referred to (Janutyte et al. 2015), though likewise not in the context of its results for upper mantle structure at the edge of the East European craton. Though the "13 BB Star" experiment used here surely has a better resolution and data quality in the region of study, the PASSEQ network also contained temporary
stations in this area and results obtained from these data should at least be mentioned in the discussion, especially if they use the same methods of analysis.

44. Added only the reference to Meier et al. 2016, since we don't use RF in the mantle any more.

The discussion should also give some idea why the Rayleigh wave dispersion curves displayed in Meier et al. (2016), which show increasing phase velocities for periods above 100 s, are so different from the one derived here, which shows decreasing velocities above 100 s, whereas the inverted velocity models contain significantly higher S-wave velocities of 5 km/s (vs. 4.7 km/s for the craton model in Meier et al.) below 150 km.

45. Please see the last paragraph of Sect. 6 of the revised manuscript.

Technical corrections There are some typings and grammar issues, i.e.
p. 1, l. 23: "a case of linearised inversion" should be "the case for linearised inversion" p. 2, l. 24: "has" should be "have"
p. 2., l. 28: "that" should be "the"
p. 4, l. 5 and l. 12: "form" should be "from"
p. 4, l. 11ff: "In average" should be "On average"
p. 7, l. 16: "to" is doubled ("analogy to to Bayesian inference")
p. 7, l. 17: brackets around "freeze" should be removed
p. 7, l. 21: brackets around references to Berteussen (1977) and Gardner et al. (1974) should be removed
p. 9, l. 1: "fit into" should be "fit the"
p. 16, l. 1ff: the reference to Ammon et al. (1990) misses one of the three authors (G. Zandt), and there's a typing error in the page number (should be 15318 instead of 153318)
p. 19, l. 7ff: "Schulte-Pelkum", and a missing space between "Bayesian" and "Monte Carlo"
Quite a few articles, both definite and indefinite, are missing, e.g. on p. 4, ll. 5-9. It might be useful to have a native speaker check the text in this respect.

46. Corrected.

References Ammon et al. (1990), JGR, 95, 15303-15318
 Fischer et al. (2010),

Ann. Rev. Earth Planet. Sci., 38, 551-575
Graw et al. (2017), BSSA, 107, 63-651, doi:10.1785/0120160262
 Dugda et al. (2007), JGR, 112, 2156-2202
Janutyte et al. (2015), Solid Earth, 6, 73-91
Jin and Gaherty (2015), GJI, 201, 1383-1398, doi:10.1093/gji/ggv079
Jones et al. (2010), Lithos, 120, 14-29, doi:10.1016/j.lithos.2010.07.013
Juliá et al. (2000), GJI, 143, 95-112
Juliá et al. (2005), GJI, 162, 555-569, doi:10.1111/j.1365-246X.2005.02685.x Kind et al. (2012), Tectonophysics, 536–537, 25–43, doi:10.1016/j.tecto.2012.03.005 Kind et al. (2015), Solid Earth, 6, 975-970, doi:10.5194/se-6-957-2015
Kind et al. (2017), Tectonophysics, 700-701, 19-31, doi:10.1016/j.tecto.2017.02.002 Knapmeyer-Endrun et al. (2017), EPSL, 458, 429-441, doi:10.1016/j.epsl.2016.11.011 Lebedev et al. (2009), Lithos, 109, 96-111, doi:10.1016/j.lithos.2008.06.010 Legendre et al. (2012), GJI, 191, 282-304, doi:10.1111/j.1365-246X.2012.05613.x Meier et al. (2016), Tectonophysiscs, 692, 58-73, doi:10.1016/j.tecto.2016.09.016 Özalaybey et al. (1997), BSSA, 87, 183-199
Rychert and Shearer (2009), Science, 324, 495-498, doi:10.1126/science.1169754 Soomro et al. (2016), GJI, 204, 517-534, doi:10.1093/gji/ggv462 Vinnik and Farra (2000), GJI, 141, 699-712
 Vinnik et al. (2015), Tectonophysics, 667, 189-198
Wathelet (2008), GRL, 35, L09301, doi:10.1029/2008GL033256

Please also note the supplement to this comment:
https://www.solid-earth-discuss.net/se-2017-58/se-2017-58-AC1-supplement.zip
* * *

---

## Author Comment (AC2) · 5 Dec 2017

The manuscript is about the joint inversion of Rayleigh wave phase velocity dispersion and P receiver function applied to 13 broadband stations that have been recording for 3 years at the south-western margin of the East European Craton. The manuscript wants to give some new input on the application of the linearized inversion, and to give constraints on the mantle structure of the study area. In my opinion this paper is too synthetic;

1. This work is meant to be a counterbalance to the prevailing number of papers which skip over the more thorough technical description of the inversion. The joint inversion

of RF and SWD is still relatively young and not yet well established method and has some inherent challenges to address which this paper tries to highlight. Furthermore, "synthetic" papers have proved to be valuable (e.g. Ammon et al. 1990), even with no field-data example (e.g. Morgan et al. 2013).

fundamental sections, like results and discussions are superficially written, while they need a longer, accurate and also descriptive argumentation, in order to demonstrate the quality and meaning of the results.

2. Please see the relevant sections in the revised manuscript. In Sect. 5 we present all the results with emphasis on the crustal model which wasn't the main goal of this study. We also describe all the inversion parameters that we used. Mantle structure (target) in turn is discussed in detail in Sect. 6.

In its actual shape the paper is poor and raw, and misses the accurate descriptions needed to deserve publication. In the following I am listing specific problems of the manuscript. 1) The results of the RF inversion only are better (although not yet fully convincing, see next point) than the results of the joint inversion of RF and SWD (Figure 11). Therefore it is not clear why the authors spend their time applying the joint inversion when inversion the RF only could give better results. If the authors want to proof that the joint inversion gives better constraints for unraveling the subsurface structure, then they have to convince the reader by adding examples (and explaining them), and with some argumentation that is lacking at the moment.

3. The data misfit is not the best measure of the result quality in this case. For joint inversion it will always be worse than for a single-data type since the former gives a trade-off between the misfit of both data types in exchange for mitigating their non-uniqueness. Please compare subfigures of Fig. 10 in the original manuscript to see how ambiguity of results of RF-based modeling (b) is reduced by including SWD data (c). This is one of the advantages of the joint inversion which has been shown in numerous other studies (see Sect. 1 for references).

2) The chosen crustal model (and "frozen") for the inversion is clearly not correct for the area. The fit between observed and synthetic RF for the initial 5 s is poor. If the paper wants to address the issue of "exploiting a priori knowledge" (as stated in the abstract), then the authors should show what happens in the inversion if the shallow part of the model is free and not "frozen", and show how their results are improved.

4. To address this problem we incorporated an intermediate step ("first-stage inversion") to adjust the crust before inverting for a deeper model (please see Subsect. "Exploitation of a priori knowledge" and Sect. "Results" in the revised manuscript). In that step we used prior knowledge about crustal structure from previous studies. The results obtained without a priori information (using homogeneous starting models, no "freezing") are attached as figures: suppl-fig08 (results), suppl-fig9-10 (misfit), suppl-fig11 (starting models). They are clearly less credible due to trapping by local minima close to the incorrect starting models.

3) In the same way in the text it is not explained how the joint inversion improves (or not improves) the results of the SWD inversion only.

5. Please see Introduction of the revised manuscript ("Compared to inversion of each of these data types alone, it provides better vertical resolution than SWD, and, unlike RF, constrains absolute shear velocities (e.g. Shen et al., 2013).").

4) The description of the results is almost lacking, it actually consists in listing the number of figures that show the results, such figures are not well described as well, and their meaning and their importance is never mentioned as well.

6. Please see rewritten section in the revised manuscript and point 2 of this response.

5) The discussion section is extremely short and it does not add anything new to previous knowledge, probably because the paper has nothing new to add to the state of the art of both the technique and structural features of the area.

7. Please see rewritten section in the revised manuscript and point 2 of this response.

As far as we know, there were no other similar studies in this area before, so the results on the mantle structure are undoubtedly a value added to the previous knowledge.

The following "promises" made in the abstract: "Several fundamental issues inherent in the linearised inversion were addressed in this work, including exploitation of a priori knowledge, choice of model's depth, trapping by local minima associated with non-uniqueness of the misfit- function optimization problem, proper weighting of data sets characterized by different uncertainties, and credibility of the final models" must be explained and discussed in this section.

8. Please see rewritten Sect. Discussion in the revised manuscript.

6) Figure4: the several RF stacks plotted on top of each other are hardly comprehensible. Each stack must be plotted singularly, for seek of clarity.

9. Corrected. Please see Fig. 4a of the revised manuscript.

7) Acronyms such as ASWMS, CPS, FWI must be explicated somewhere in the text

10. Corrected.

8) Figures 10, 11, and 12 deserve a complete caption; the colors in these figures are not explained at all

11. Corrected. Please see Fig. 12-15 of the revised manuscript.

Technical corrections: Page 1 line 4: linearised → linearized Page 2, line 9: covering THE entire Page 2 lines 26-28: the sentence is badly written, and should be rewritten Page 8 Line 3: cover → covers

12. Corrected.

13. References

Ammon et al. 1990, On the Nonuniqueness of Receiver Function Inversions, Journal of Geophysical Research

Morgan et al. 2013, Next-generation seismic experiments: Wide-angle, multi-azimuth, three-dimensional, full-waveform inversion, Geophysical Journal International

Please also note the supplement to this comment:
https://www.solid-earth-discuss.net/se-2017-58/se-2017-58-AC2-supplement.zip